# Microenvironment-derived factors driving metastatic plasticity in melanoma

Isabella S. Kim[1,*], Silja Heilmann[2,†,*], Emily R. Kansler[1,3,*], Yan Zhang[1], Milena Zimmer[1], Kajan Ratnakumar[1], Robert L. Bowman[1,3], Theresa Simon-Vermot[1], Myles Fennell[1], Ralph Garippa[1], Liang Lu[4], William Lee[2], Travis Hollmann[5], Joao B. Xavier[2] & Richard M. White[1,6]

Cellular plasticity is a state in which cancer cells exist along a reversible phenotypic spectrum, and underlies key traits such as drug resistance and metastasis. Melanoma plasticity is linked to phenotype switching, where the microenvironment induces switches between invasive/ $MITF^{LO}$ versus proliferative/ $MITF^{HI}$ states. Since MITF also induces pigmentation, we hypothesize that macrometastatic success should be favoured by microenvironments that induce a $MITF^{HI}$/differentiated/proliferative state. Zebrafish imaging demonstrates that after extravasation, melanoma cells become pigmented and enact a gene expression program of melanocyte differentiation. We screened for microenvironmental factors leading to phenotype switching, and find that EDN3 induces a state that is both proliferative and differentiated. CRISPR-mediated inactivation of EDN3, or its synthetic enzyme ECE2, from the microenvironment abrogates phenotype switching and increases animal survival. These results demonstrate that after metastatic dissemination, the microenvironment provides signals to promote phenotype switching and provide proof that targeting tumour cell plasticity is a viable therapeutic opportunity.

[1] Memorial Sloan Kettering Cancer Center, Department of Cancer Biology & Genetics, New York, New York 10065, USA. [2] Memorial Sloan Kettering Cancer Center, Department of Computational Biology, New York, New York 10065, USA. [3] Memorial Sloan Kettering Cancer Center, Gerstner Graduate School of Biomedical Sciences, New York, New York 10065, USA. [4] Medical College of Georgia, Augusta, Georgia 30912, USA. [5] Memorial Sloan Kettering Cancer Center, Department of Pathology, New York, New York 10065, USA. [6] Memorial Sloan Kettering Cancer Center, Department of Medicine, New York, New York 10065, USA. † Present address: The Danish Stem Cell Centre (DanStem), University of Copenhagen, 2200 Copenhagen N, Denmark. * These authors contributed equally to this work. Correspondence and requests for materials should be addressed to R.M.W. (email: whiter@mskcc.org).

Prevailing models of metastasis have implicated changes in cellular differentiation as a key driver of the process. These models have suggested that undifferentiated/invasive cells initiate metastasis, and these cells then seed secondary organs to establish successful macrometastases. Two models for cell state transitions in melanoma have been proposed: the cancer stem cell model (hierarchical, irreversible) and the phenotype switching/plasticity model (reversible). An unanswered question prevails in the existing models: how do the slowly proliferative, invasive cells which arrive at new sites exit that state to then resume proliferation in secondary sites? In melanoma, these changes in cell state are regulated by the transcription factor MITF, which has been proposed to act as a molecular rheostat: cells with low levels of MITF are invasive, whereas those with higher levels of MITF are proliferative[1–3]. Because MITF also drives differentiation, one implication of this model is that cells which are ultimately successful in metastatic sites should be simultaneously proliferative and differentiated. As switches between the invasive/MITF$^{LO}$ versus proliferative/MITF$^{HI}$ states are posited to be due to microenvironmental factors, this would paradoxically imply that microenvironments which promote the differentiated cell state would be most strongly associated with metastatic success[4]. The identity of such differentiation inducing factors in melanoma metastasis remains largely unknown, yet has important consequences in understanding the forces that drive macrometastatic colonization. To address this idea, we have used a zebrafish model of melanoma to monitor changes in cell differentiation during metastatic engraftment, and find that although early metastases are seeded by undifferentiated cells, over time these cells enact a gene program of melanocytic differentiation that is strongly associated with proliferation. The microenvironment induces this differentiated metastatic state in part through the developmental morphogen EDN3. Prevention of this differentiated cell state via EDN3/ECE2 inactivation doubles survival of the animals, suggesting that the acquisition of a differentiated cell state is necessary for metastatic success in melanoma, and can be prevented by interrupting microenvironment-melanoma cross-talk. Our data would imply that solely targeting undifferentiated cell populations is likely to miss an important component of metastatic lesions.

## Results

**Zebrafish imaging demonstrates that metastases differentiate.** We utilized a zebrafish model of melanoma to monitor changes in differentiation during metastatic spread. We created transgenic zebrafish in which the melanocyte-specific mitfa promoter drives the human BRAF$^{V600E}$ gene[5,6] along with a mitfa-green fluorescent protein (GFP) reporter cassette. From this animal, we derived a zebrafish-specific melanoma cell line, ZMEL1-GFP, which can be transplanted into the transparent casper strain of fish and metastatic patterns visualized using in vivo imaging[7]. Similar to most human melanoma cell lines, the ZMEL1-GFP line was completely unpigmented in vitro and has a highly mesenchymal appearance (Fig. 1a, left). This allowed us to monitor differentiation using both the mitfa-GFP transgene as well as melanin pigmentation of the tumours. Whereas melanin in mammals is not a consistent marker of differentiation, zebrafish produce a very dark variant of melanin that is easily visualized, because fish carry a polymorphism of the pigmentation gene SLC24A5 that is typically associated with darker skin in individuals of African descent[8]. Upon transplantation into subcutaneous sites (Fig. 1a, top), akin to an in-transit metastasis in humans, we first observed that local tumours were mitf-GFP + but completely

unpigmented. With a latency of 7–14 days, 100% of the locally engrafted skin tumours regained pigmentation, a reflection of melanocytic differentiation. The secondary subcutaneous metastases that then developed were also initially mitf-GFP + and unpigmented, but 53% of the subcutaneous metastases then became visibly pigmented by day 14 (Fig. 1a and Supplementary Movies 1 and 2). This suggested that metastases are initially undifferentiated but became differentiated after engraftment. To confirm this, we directly transplanted ~50 unpigmented ZMEL1-GFP cells into the vasculature of larval casper zebrafish, bypassing the initial skin site (Fig. 1a, bottom). We used larvae for this assay, rather than adults, due to the greater accessibility of the vasculature. Cells initially circulated widely, but within 24 hours extravasated in several areas including the skin, caudal haematopoietic territory and the eye. Over the next 7–28 days, these animals formed widespread macrometastases that were also visibly pigmented (Fig. 1a, bottom), the appearance of which is dominated by the superficial subcutaneous metastases. In both the adult and larval assays, this pattern of metastatic spread is analogous to Stage IV (Tx/Nx/M1a) disease. Histological analysis of a widely disseminated tumour confirmed clear areas of pigmentation in muscle invasive disease (Fig. 1c) adjacent to the overlying skin. To ensure that our findings were not unique to the ZMEL1 line, we performed similar experiments with an additional zebrafish melanoma cell line, ZCREST1. This line is derived from a BRAF;p53 tumour but the reporter GFP gene is driven by the crestin neural crest reporter, as previously described[9]. Similar to the to the ZMEL1 results, the initially engrafted ZCREST tumours were GFP + but unpigmented, but had become pigmented by day 14–44 post-transplant (Supplementary Fig. 2). These data are consistent with mouse studies showing acquisition of a pigmented phenotype after dissemination and engraftment[10]. Pigmentation of melanocytes typically coincides with the acquisition of a dendritic phenotype[11]. To further confirm that the initially undifferentiated ZMEL1-GFP cells could actively differentiate in vivo, we used time-lapse microscopy of cells injected into the vasculature of the fish to analyse their morphology. Within 57 hours post extravasation, the melanoma cells acquire a highly dendritic appearance characteristic of differentiated melanocytes, and by 5 days post transplant, 83% had acquired a dendritic appearance (defined as greater than two projections away from the cell body) compared with none on the day of transplant (Fig. 1c,d, Supplementary Fig. 1 for quantification of dendricity and Supplementary Movie 3 for time-lapse imaging movie), confirming that post-extravasation differentiation is an early step in metastatic engraftment.

**Metastases exhibit a differentiation signature.** To quantify the extent of this differentiation program, we performed RNA-seq on the metastatic ZMEL1-GFP cells as compared with the parental cells in vitro, generating a list of ~2,000 genes that were altered after dissemination (Fig. 2a and Supplementary Data 1). We found that canonical differentiation genes such as PMEL, TYR, TYRP1 and DCT were all upregulated in the disseminated cells (Fig. 2b), all known targets of MITF[12]. We found only a small increase in MITFB and no increase in MITFA (the two zebrafish orthologues of MITF), suggesting that it is increased activity, rather than expression, of MITF that was leading to the differentiation program. We also found a significant increase in markers of proliferation such as MYC and MYCL (Supplementary Fig. 3), consistent with prior observations that differentiation and proliferation are tightly linked in melanoma[2], in part by regulating levels of MITF activity[13]. We used Gene set enrichment analysis (GSEA) to compare the zebrafish

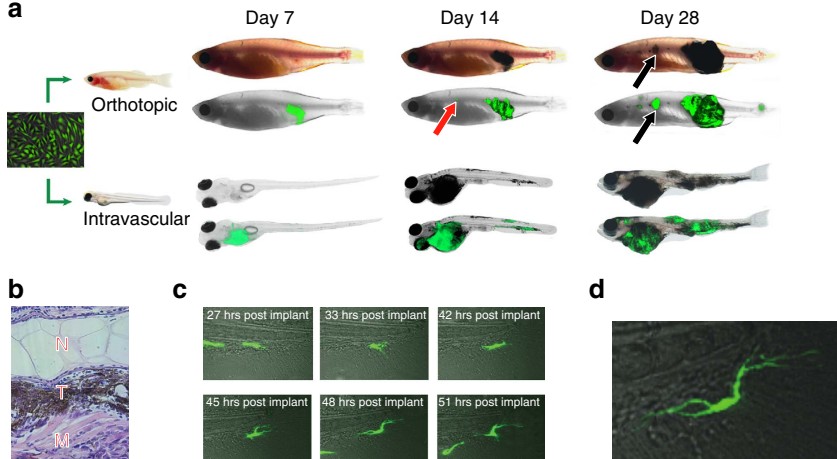

**Figure 1 | Melanomas become differentiated after dissemination to secondary sites.** (**a**, top) ZMEL1-GFP zebrafish melanoma cells (mitf-BRAF$^{V600E}$; p53$^{-/-}$) were orthotopically transplanted into the ventral skin of the transparent *casper* strain and then imaged using brightfield and GFP over 28 days. At days 1 and 7 post-transplant, the tumours are GFP+ but devoid of pigmentation, a marker of differentiation. By day 14 post transplant, the primary tumour mass in the orthotopic site has become deeply pigmented and the animal has developed small anterior metastases that are GFP+ but unpigmented (red arrowhead). By day 28, these metastases have now enlarged and are clearly pigmented (black arrowhead), consistent with metastatic differentiation. (**a**, bottom) ZMEL1-GFP cells were transplanted into the vasculature of larval *casper* recipients to assess direct differentiation capacity at sites of metastatic colonization, bypassing the primary skin site. Similar to what is seen in the orthotopic transplantation, the cells are initially unpigmented at days 1 and 7, but become increasingly pigmented at days 14 and day 28, indicating that cells can directly differentiate after extravasation. (**b**) Histological analysis of a larval *casper* recipient transplanted with ZMEL1-GFP cells shows heterogeneous acquisition of pigmentation, with cells near muscle invasive disease showing increased evidence of melanization (N = notochord, T = tumour, M = muscle). (**c**) Time-lapse imaging of ZMEL1-GFP morphology in a larval *casper* recipient shows that cells that exit the vasculature and enter the tailfin epithelial layer gradually acquire a dendritic phenotype that is characteristic of differentiated melanocytes. (**d**) Enlargement of a dendritic cell in the tailfin. Images are representative of n = 10–20 fish per group.

metastatic signature to a panel of known human differentiation genes (from ES-derived melanocytes)[12]. We found a significant enrichment between the disseminated zebrafish cells and human differentiated melanocyte genes (NES = 1.276, FDR = 0.009), including PMEL, TYR and TYRP1 (Fig. 2c and Supplementary Data 2). We also used GSEA to compare our disseminated ZMEL1-GFP signature to a panel of human melanomas that have been previously classified as 'invasive' versus 'proliferative'[14], and found that the metastatic zebrafish cells were strongly enriched for the 'proliferative' signature that is dominated by differentiation-related target genes of MITF such as PMEL, TYR, SLC45A2 and GPR143, (Fig. 2c and Supplementary Data 3, NES = 1.317, FDR = 0.08). Taken together, this data demonstrates that successful macrometastatic engraftment is associated with the acquisition of a differentiated and proliferative cell state.

**Differentiation genes predict worse outcome in humans.** This data suggests that the gene programs associated with metastatic cells phenotypically switching to a more differentiated state should predict prognosis in human metastatic melanoma. To examine this question, we generated a list of 83 differentiation-associated genes (that is, PMEL, TYR, TYRP1, DCT, SLC45A2, MLPHA, MLANA) derived from the GSEA analysis above as well as the ZFIN database (http://zfin.org/), and then assessed how their expression correlated with survival in the TCGA melanoma cohort. We stratified patients by those who presented with stage I (localized) versus stage III/IV (lymph node or distant metastases). In those presenting with stage I disease (even if they eventually went on to develop metastases, as did most patients in the TCGA cohort), none of the differentiation associated genes were associated with prognosis (Fig. 2e and Supplementary Figs 4 and 5), which is consistent with prior reports indicating no relationship between pigmentation of primary tumours with prognosis[15]. In

contrast, in stage III/IV disease, a cluster of genes indicative of differentiation state all portended a significantly worse prognosis: PMEL, EDNRB, TYR, GPR143, SLC45A2, MLPH and BACE2 (Fig. 2f and Supplementary Figs 4 and 5). Examination of the Kaplan–Meier curves for the most strongly associated gene, PMEL (Fig. 2g,h), shows that patients with the highest level expression, and not intermediate or low levels, had the worst prognosis. This extends to other differentiation associated genes such as TYR and TYRP1 (Supplementary Fig. 6). This data indicates that these pigmentation genes stratify those who present with advanced stage III/IV disease and is associated with macrometastatic growth once the cell has arrived at a secondary location.

**Microenvironmental factors promoting plasticity.** We wanted to identify the microenvironmental triggers of this differentiation program. We analysed our RNA-seq data using Ingenuity Pathway Analysis (IPA), including the canonical and upstream regulatory analyses. IPA makes statistical predictions (z-scores) of what upstream molecules or pathways are likely to have produced that gene signature. From this analysis, we identified 253 potential molecules that could yield the disseminated melanoma signature, which we then categorized into six pathways that we thought likely activated by extracellular/secreted factors: endothelin, dopamine, IGF, cholesterol, fatty acid and c-kit signalling (Fig. 3a, left panel). We predicted that if we applied agonists of these pathways to our cells *in vitro*, we would recapitulate the differentiated/proliferative state we had seen *in vivo*. Using the InCell6000 high-throughput/high-content imaging system, we performed a small molecule *in vitro* screen (Fig. 3b, middle and right panels and Supplementary Data 4) to identify molecules that could induce both differentiation (read out by elongation morphology) as well as proliferation (read out by cell number). We tested these in the zebrafish ZMEL1 melanoma cells, along with

human A375 and SK-Mel28 melanoma cells to ensure that the effects we see are not fish-specific. In the ZMEL1 line (Fig. 3b), we found that 5/7 tested molecules (EDN1, EDN3, IGF1, SCF and LDL-cholesterol) significantly increased proliferation, whereas

dopamine/L-DOPA caused a marked decrease in cell proliferation. In contrast, we found that only EDN1, EDN3 and Dopamine/L-DOPA caused a significant elongation phenotype (Fig. 3d), consistent with differentiation. To orthogonally confirm

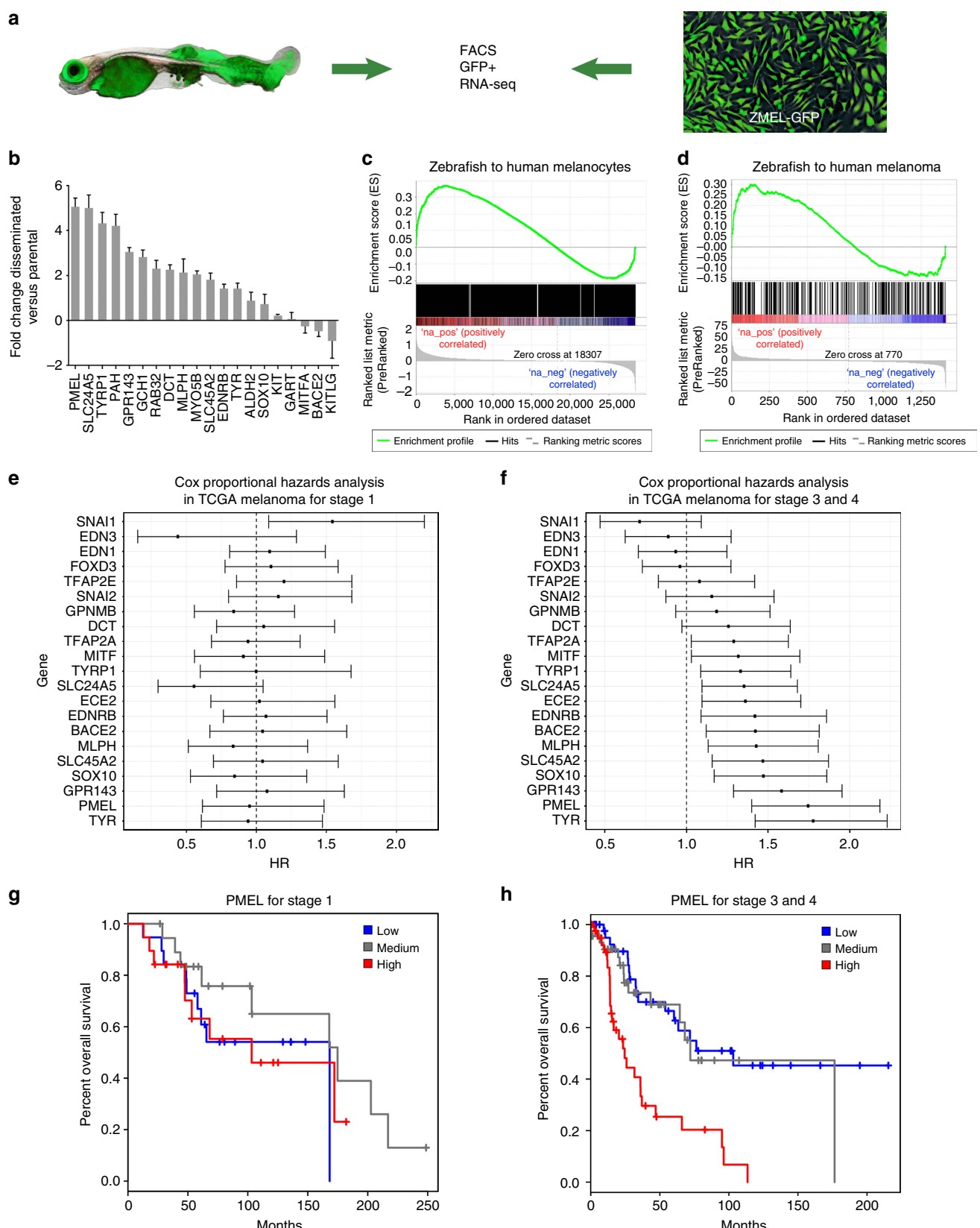

that the elongation morphology represented bona fide differentiation, we treated ZMEL1-GFP cells with either EDN3 or L-DOPA and then directly measured melanin content using spectrophotometric absorbance (Fig. 3f) and examination of the cell pellets (Fig. 3g). This confirmed that both of these molecules significantly increase melanin content/differentiation. In the human cells, we found that EDN3 caused a significant increase in both proliferation and elongation in A375 (Fig. 3c,e) and SK-Mel28 (Supplementary Fig. 7) cells, whereas dopamine/ L-DOPA only caused a block in proliferation and did not cause the elongation phenotype that was seen in the fish cells. When taken as a whole, this data suggests that melanoma cells use a complex set of interconnected signals to enact the differentiated/proliferative state. This state relies on high levels of endothelins/SCF/IGF1 to promote proliferation, but low levels of dopamine/L-DOPA to allow for differentiation but not at a concentration that would adversely affect proliferation (Fig. 3h). Among these various factors, our data indicate that microenvironmental EDN3 is uniquely capable of inducing a phenotype switch to yield melanoma cells that are both proliferative and differentiated across both zebrafish and human conditions.

**Loss of microenvironmental EDN3/ECE2 abrogates phenotype switching.** Endothelins such as EDN3 are vasoactive peptides that are expressed in the zebrafish epidermis[16], likely by the keratinocytes which normally surround the melanocytes[17,18]. The endothelins are known to act as key inducers of normal embryonic and adult melanocyte formation by binding to the EDNRB receptor[19] and have been previously implicated in melanoma growth by some but not all groups[20–22]. We hypothesized that inactivation of EDN3 in the microenvironment would abrogate phenotype switching in the nearby melanomas. To test this, we created CRISPR mutant-recipient zebrafish in which we introduced loss-of-function mutations in EDN3b, one of the two zebrafish orthologues of EDN3. In addition, we also created CRISPR mutants against endothelin-converting enzyme 2, ECE2b. This is the proteolytic enzyme recently shown to synthesize EDN3b in zebrafish[16] and has been shown to be one of the enzymes involved in endothelin synthesis in keratinocytes[23]. ECE2 was also one of the differentiation-associated genes we found associated with worse prognosis in human patients. We performed these experiments using the AB strain of zebrafish, rather than *casper,* because we wished to ascertain the strength of the EDN3b or ECE2b mutation itself on normal melanocyte development, and to ensure that the CRISPR was leading to a true loss of function (Supplementary Fig. 8). We found that melanocyte development in both the EDN3b mutant and ECE2b mutant was severely abnormal (Fig. 4a–c), with loss of the dorsal and ventral melanocyte stripe, an overall reduction in the number of mature pigment cells, and a misshapen appearance of the residual melanocytes. This is consistent with a role for EDN3b and ECE2b in promoting melanocyte differentiation and

growth[16,24,25]. We then transplanted ZMEL1-GFP cells subcutaneously into either WT, EDN3b or ECE2b-deficient recipients to ascertain the microenvironmental influence on melanoma proliferation and differentiation (Fig. 4d). Because in the wild-type (WT) background, we cannot reliably quantify distant metastases, we transplanted the ZMEL1-GFP sites into multiple subcutaneous sites, essentially to mimic a patient with multiple subcutaneous metastases. Melanomas that developed in both the EDN3b or ECE2b-deficient backgrounds were significantly smaller than those that developed in control animals (Fig. 4e–g, quantified in 4h), suggesting that EDN3b supports melanoma proliferation in this site. Moreover, the tumours that developed in the EDN3b or ECE2b-deficient backgrounds were markedly less pigmented with tumours typically having a greyish appearance, and exhibited less mitf-GFP fluorescence compared with WT recipients (Fig. 4i), consistent with lack of differentiation. These phenotypic effects translated to a significant improvement in survival of the animals (Fig. 4j). At 50 days post-transplant, we found that only 6.25% of the WT animals had survived, in contrast to 42.1% of the ECE2b and 54.1% of the EDN3b-recipient animals ($P = 0.0035$, log-rank test). To further ensure that these results were not unique to the AB strain, we performed additional transplants in the *casper* strain as well, and found that the EDN3b-deficient recipients similarly developed smaller tumours that were less pigmented (Supplementary Fig. 9).

We quantified the differences in the tumours in the WT versus EDN3b versus ECE2b backgrounds using histologic sectioning of tumours (Fig. 5, $n = 4$ tumours from each genotype). By haematoxylin and eosin staining, we found that the tumours in both mutant backgrounds had clear evidence of central necrosis ($n = 4/4$ tumours for both EDN3b and ECE2b, $n = 0/4$ for WT tumours). This was accompanied by a significant increase in the number of cleaved caspase positive cells in both mutant backgrounds (WT $= 2.05 + / - 0.35$, ECE2b $= 3.35 + / - 0.34$, EDN3b $= 3.7 + / - 0.41$ mean cells per hpf $+ / -$ s.e.m., $P < 0.05$, ANOVA). We found a modest decrease in proliferation as marked by pH3 staining in the EDN3b background (WT $= 11.8 + / - 0.93$, EDN3b $= 8.2 + / - 0.5$ mean cells per hpf $+ / -$ s.e.m., $P < 0.05$, ANOVA). Confirming the *in vivo* brightfield imaging results, we saw a significant decrease in melanin content, as assessed by Fontana-Masson staining, in both mutant backgrounds (WT $= 1143 + / - 186$, ECE2b $= 382 + / - 140$, EDN3b $= 324 + / - 48$ area units per hpf $+ / -$ s.e.m., $P < 0.05$, ANOVA). In all three tumour types, the tumours cells stained uniformly for both GFP and BRAF$^{V600E}$, indicating that the smaller, less pigmented tumours in the mutant backgrounds were not due to silencing of the transgenes.

Collectively, these data demonstrate that microenvironmental EDN3b, acting via ECE2b, promotes phenotype switching of melanoma cells by simultaneously inducing a cell capable of both differentiation and proliferation. We find that abrogation of phenotype switching in the melanoma makes the tumours ultimately less lethal to the animal.

**Figure 2 | Disseminated metastatic cells exhibit a differentiation gene signature.** (**a**) ZMEL1-GFP cells were transplanted into the vasculature of a larval zebrafish, and then fish grew until 21 days when they had widespread tumour dissemination. These fish were then disaggregated and the post-dissemination ZMEL1-GFP + cells were isolated by fluorescence-activated cell sorting (FACS) (left). The parental ZMEL1-GFP cells maintained in culture were trypsinized and similarly subject to FACS sorting (right). These two populations were then subject to RNA-seq. (**b**) Expression of melanocyte/ pigmentation genes (that is, PMEI, TYR, SLC24A5) in the disseminated ZMEL1 cells compared with parental, showing a significant upregulation of a differentiation gene program. (**c**) GSEA shows a significant enrichment between the disseminated ZMEL signature and a signature of human differentiated melanocytes. (**d**) GSEA shows a significant enrichment between the disseminated ZMEL1 signature and the human melanoma subtypes classified as 'differentiated/proliferative'. (**e,f**) Cox proportional Hazard model of TCGA data for melanocyte differentiation genes in either stage I (**e**) localized disease versus stage III/IV (**f**) metastatic disease shows that the differentiation signature portends a worse prognosis in metastatic patients. (**g,h**) Kaplan–Meier survival analysis for the melanosome protein PMEL, showing significantly worse survival for stage III/IV patients compared with stage I. Error bars are s.e.m., with $n = 2$–6 animals per group.

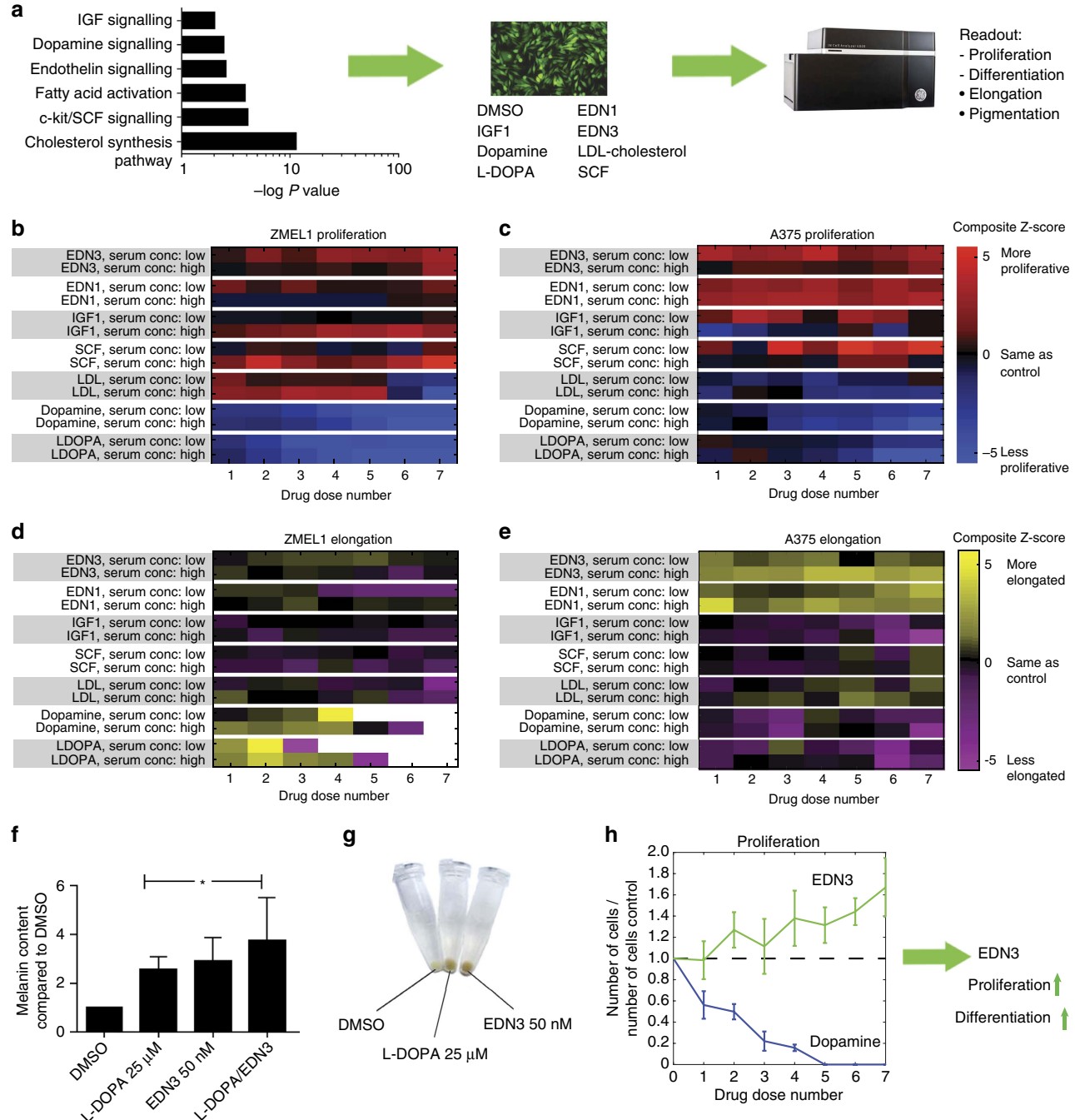

**Figure 3 | Microenvironmental factors inducing the differentiated/proliferative state.** (**a**) Ingenuity Pathway Analysis of ZMEL1-GFP cells after metastatic dissemination suggests six pathways (left) that could mediate microenvironmental-mediated differentiation/proliferation. *P* values indicate estimated likelihood that the indicated pathway is altered in the RNA-seq data set. To test these pathways, ZMEL-GFP cells were treated *in vitro* with agonists of these six pathways at the indicated doses (middle) and differentiation and proliferation was assessed using the InCell high-throughput/high content scanning system (right). (**b**,**c**) Heatmaps showing the effect of the various agonists on proliferation tested in either low- or high-serum conditions in ZMEL1 melanoma cells (**b**) or A375 melanoma cells (**c**). Red indicates increased proliferation, whereas blue indicates decreased proliferation (compared with dimethylsulphoxide control). (**d**,**e**) Heatmaps showing the effect of the agonists on cell elongation, which is a reflection of melanoma differentiation, in ZMEL1 (**d**) or A375 (**e**) melanoma cells. Yellow indicates increased elongation, while purple indicates decreased proliferation. (**f**) To more directly test whether elongation is associated with differentiation, ZMEL1-GFP cells were treated with Endothelin-3 and L-DOPA, or the combination and melanin content assessed. Both agonists caused a significant increase in melanin content (* = *P* < 0.05). (**g**) Photograph of the cell pellets (equal numbers of cells) treated with either dimethylsulphoxide, Endothelin-3 or L-DOPA. (**h**) Proliferation curves showing opposing effects of Endothelin-3 as compared with L-DOPA in ZMEL1 cells, suggesting that EDN3 is uniquely capable of inducing both proliferation and differentiation across multiple cell types. Error bars are s.e.m., with *n* = 4–6 biological replicates per group.

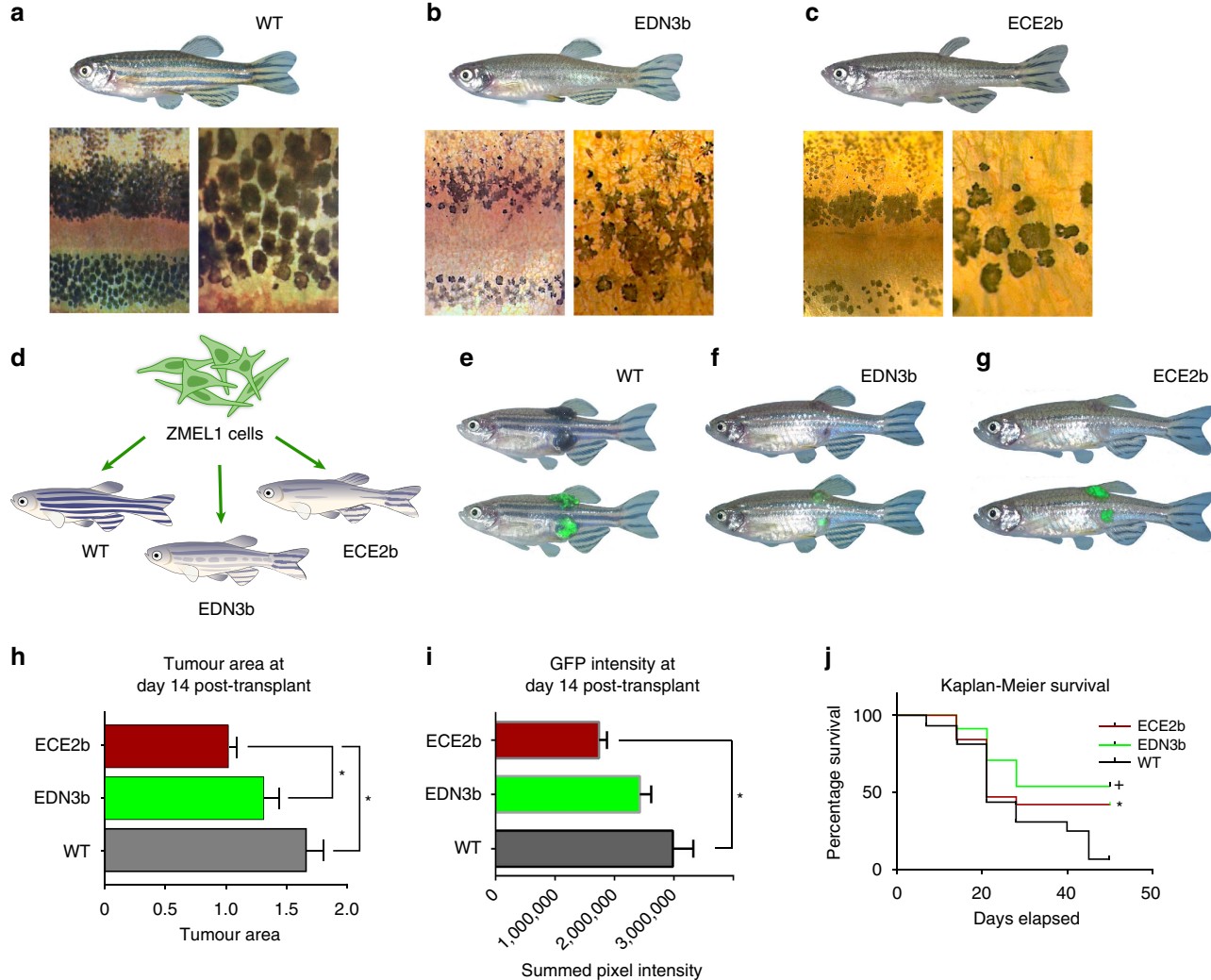

**Figure 4 | Microenvironmental CRISPR against EDN3b abrogates phenotype switching.** (**a**) Control wild-type fish of the AB strain that was injected with Cas9 protein alone. Higher magnification views of the melanocytes from this fish is shown below, indicating these are fully mature, pigmented melanocytes. (**b**) An AB fish in the F1 generation that had been injected with a CRISPR gRNA against EDN3b, showing a severe loss of melanocytes over the entire body of the zebrafish. Higher magnification views (below) from this fish show a decreased number of pale, misshapen melanocytes. (**c**) An AB fish in the F1 generation that had been injected with a CRISPR gRNA against ECE2b, showing a melanocyte defect that is highly similar to that seen in the EDN3b mutant. (**d**) Schema for testing the effects of microenvironmental EDN3b on melanoma growth and differentiation. Equal numbers of ZMEL1-GFP cells were transplanted subcutaneously either in a WT recipient, EDN3b mutant recipient or the ECE2b mutant recipient, who differ only in the loss of function EDN3b in the microenvironment. (**e**) Tumour growth in the WT recipient shows large pigmented tumours in multiple subcutaneous sites, consistent with phenotype switching to a more differentiated/proliferative state. (**f,g**) Tumour growth in the EDN3b or ECE2b recipients is impaired, with smaller tumours that are markedly less pigmented, consistent with reduced phenotypic switching due to loss of endothelin-3 signalling from the microenvironment. (**h**) Quantification of tumour area in WT versus EDN3b versus ECE2b backgrounds at 14 days post transplant demonstrates a significant decrease in tumour size in the CRISPR mutant (* = $P < 0.05$, WT versus EDN3b and WT versus ECE2b, ANOVA, $n = 15$ WT, $n = 24$ EDN3b, $n = 19$ ECE2b). (**i**) Measurement of mitf-GFP pixel intensity in WT versus EDN3b versus ECE2b recipients shows a significant decrease in overall GFP + intensity in the ECE2b mutant (* = $P < 0.05$, WT versus ECE2b). (**j**) Kaplan–Meier survival curves of WT versus EDN3b $-/-$ recipient fish, showing a significantly longer survival time in the EDN3b and ECE2b recipients (+, $P = 0.0035$, logrank test). Error bars are s.e.m. with the numbers of animals indicated above.

## Discussion

The capacity for cells to successfully colonize secondary organs during metastatic spread has a strong temporal component. Cells that initially arrive at new sites are invasive, yet poorly proliferative, since these two programs tend to exist at opposite ends of a phenotype spectrum[26]. After initial extravasation, these nascent metastatic cells must inevitably exit this invasive state and begin to proliferate again, but how cells make that transition in melanoma has remained elusive. Our data would suggest that the reacquisition of a proliferative state in metastatic sites is tightly linked to the acquisition of a differentiation gene program. This likely reflects, in part, the dual capacity of the melanocyte master regulator MITF to induce both proliferation and differentiation. In support of this idea, we find that zebrafish melanomas harbour clear characteristics of a differentiated, pigmented cell state after extravasation. Most importantly, we find that human metastatic tumours that harbour a similar differentiation gene signature portend a much worse prognosis, likely because those tumours are more capable of metastatic proliferation.

We have identified multiple microenvironmental factors inducing this differentiation gene signature. Our data

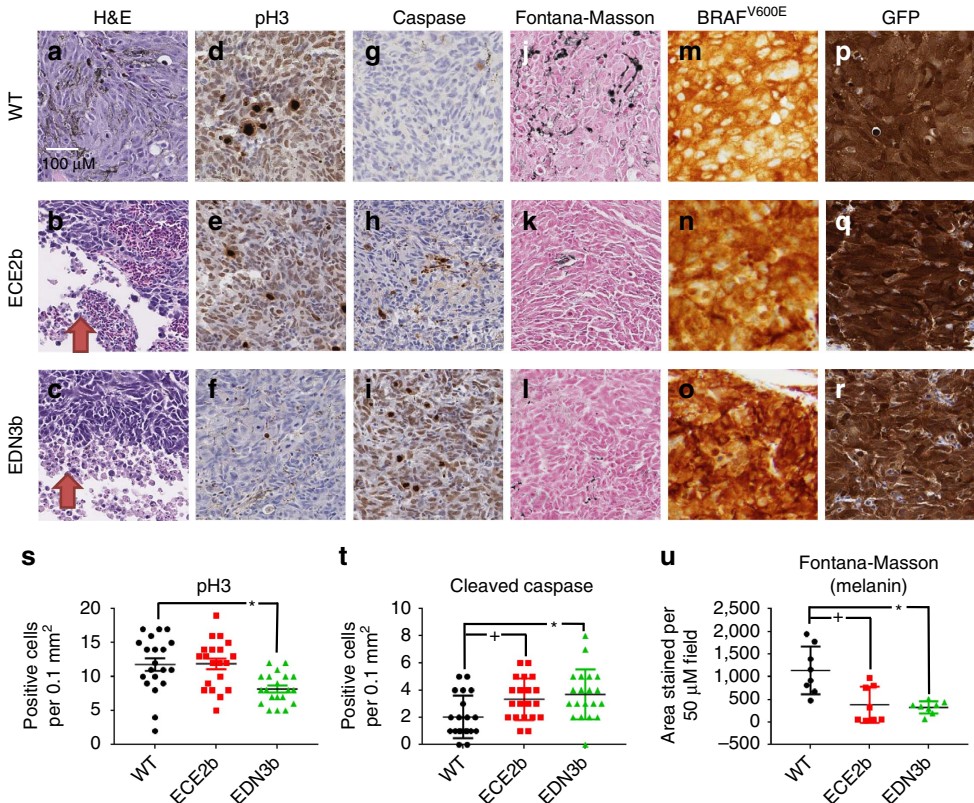

**Figure 5 | Histologic analysis of melanomas that develop in WT background versus ECE2b or EDN3b-deficient backgrounds.** (**a**–**c**) H&E staining demonstrates areas of central necrosis (red arrows) in the ECE2b and EDN3b-deficient background when compared with WT (WT = 0/4, ECE2b = 4/4, EDN3b = 4/4 with central necrosis). (**d**–**f**) A significant decrease in proliferation, as assessed by phospho-H3 staining, is seen in the EDN3b-deficient tumours, which is quantified in the (**s**) panel below (*$P < 0.05$, ANOVA). (**g**–**i**) A significant increase in apoptosis as measured by cleaved caspase expression is seen in both ECE2b and EDN3b-deficient backgrounds, quantified in panel (**t**) below (*, +, $P < 0.05$, ANOVA). (**j**–**l**) A significant decrease in melanin content, as measured by Fontana-Masson staining, is seen in the ECE2b and EDN3b backgrounds as compared with WT and quantified in panel (**u**) below (*, +, $P < 0.05$, ANOVA). (**m**–**o**) Staining for human BRAF$^{V600E}$ was uniform across all three genotypes, as was GFP expression (**p**–**r**), indicating that expression of those transgenes from the mitfa promoter was not affected. Error bars are s.e.m.

demonstrate a unique role for EDN3, and its synthetic enzyme ECE2, as phenotype switching factors in the microenvironment. It is one factor that can promote a cell state capable of both proliferation and differentiation, which strongly promotes metastatic outgrowth. Recent studies have implicated enhanced endothelin signalling in the propensity of melanoma to metastasize to the brain, and EDN3 overexpression in mice leads to more metastatic and hyperpigmented melanomas[27]. This may be in part due to increased phenotype switching towards a differentiated state. The mechanism by which lack of microenvironmental endothelins lead to smaller, less pigmented tumours is likely due to a direct effect on tumour cells (via the EDNRB receptor on the melanoma cell) in addition to possible vascular effects from its role in endothelial cells (as evidenced by central necrosis in our tumours). However, clinical trials of endothelin antagonists have thus far been disappointing[28]. In part, this may be due to the relatively low potency of EDNRB-specific antagonists. We tested one such antagonist, IRL2500 (ref. 29), and found that it only inhibited melanoma growth at high concentrations that are unlikely to be clinically attainable (Supplementary Fig. 10). The development of more potent EDNRB antagonists are being developed[30] and may yield benefit in the future but there are at least two other potential reasons for limited clinical efficacy thus far. First, EDN3 likely exists as part of a network, along with IGF1, SCF, and dopamine, to exert phenotype switching in metastatic sites (schematically

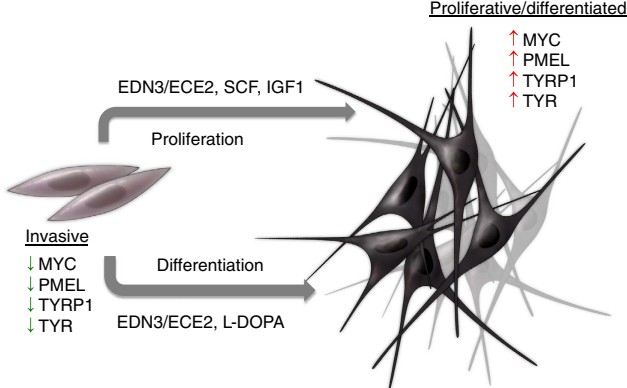

**Figure 6 | An overview model of phenotype switching factors in melanoma.** The major driver of both proliferation and differentiation is EDN3 (synthesized by ECE2b), whereas proliferation alone can be augmented by pro-proliferation signals such as SCF and IGF1 and differentiation augmented by L-DOPA. These convergent signals yield cells that transition from the invasive gene signature to an MITF-dominated gene signature with expression of proliferation genes (MYC) and differentiation genes (PMEL, TYRP1, TYR).

illustrated in Fig. 6). For example, we find that dopamine (an intermediate in the synthesis of melanin) may promote differentiation, and a recent study using labelled betaCIT, a cocaine derivative that reflects the dopamine uptake transporter[31], inadvertently identified an undiagnosed melanoma brain metastasis. The plasma ratio of L-DOPA to tyrosine is associated with increased metastatic propensity[32]. The mechanism by which L-DOPA leads to enhanced differentiation (at the cost of proliferation) remains to be determined in future studies, but may be related to its known role in synthesis of melanin itself since it is the substrate for the TYR gene product, tyrosinase. Second, while prevention of phenotype switching towards differentiation may counteract macrometastatic growth, it may paradoxically leave behind a pool of highly invasive melanoma cells untouched, with subsequent negative consequences for the patient. Recent work has identified Wnt/ROR kinase and TGFβ signalling as potential microenvironmental supporters of this invasive state[10,33]. We would predict that simultaneous targeting of the microenvironmental forces that promote both the invasive (via Wnt/TFGβ) and differentiated (via EDN3/IGF/SCF/ dopamine) cell states will be required for long term efficacy, a concept which remains to be explored in future studies.

## Methods

**Cell culture.** All cells were maintained in DMEM with 10% FBS/pencillin/strep-tomycin/glutamine. The ZMEL1-GFP cells were derived as previously described[7] and came from a transgenic zebrafish with the mitf-BRAF$^{V600E}$ and p53 − / − background. The A375 and SKMel28 cells were obtained directly from the American Type Culture Collection.

**Adult transplantation.** The zebrafish melanoma cell line ZMEL1-GFP was used as the parental population. This line contains a transgene in which the mitfa promoter drives human BRAF$^{V600E}$ and also harbours a p53$^{−/−}$ loss of function mutation. The GFP reporter cassette in the parental line is also driven by the mita promoter. For the transplantation, adult *casper*-recipient zebrafish were irradiated with a total of 30 Gy (15 Gy split dose on days − 4 and − 3 before transplant). In all, 300,000–500,000 cells were resuspended in 1–5 μl of 0.9X PBS, and then transplanted into either the ventral or dorsal skin (using a pulled glass micropipette). The fish were imaged within 1d to ensure transplantation success, and then serially imaged over a period of 30 days using brightfield and fluorescence imaging.

**Larval transplantation.** Adult *casper* fish were incrossed, and embryos were cleaned and dechorionated. At 2 days post fertilization, approximately 50 ZMEL1 cells resuspended in 1nl of 0.9 × PBS were directly transplanted into the circulation via the Duct of Cuvier technique. The fish were imaged immediately after imaging and any fish that did not receive the proper number of cells was discarded. The fish were imaged as above until approximately day 30 post transplant. No irradiation was used in this assay.

**Imaging and image processing.** Fish were imaged using an upright Zeiss Discovery V16 equipped with a motorized stage, brightfield, GFP and tdTomato filter sets. To acquire images, fish were lightly anaesthetized with Tricaine 4 mg ml$^{−1}$. Images were acquired with the Zeiss Zen software v1, and the post image processing was done using MATLAB or ImageJ. Quantification of pigmentation in the larval stage images was done using brightfield imaging after thresholding for black versus white areas of the fish, and then calculated as the percentage area of the tail occupied by black pigmented cells, see paragraph further down for quantification of pigmentation in adult fish images.

**Histology.** Selected fish were fixed in 4% PFA and then paraffin embedded. Fish were sectioned at 5 uM, and then stained with H&E, phospho-H3, cleaved caspase, Fontana-Masson, anti-GFP or anti-BRAV600E. All histology was performed by Histowiz (http://www.histowiz.com) and reviewed and quantified by pathologist (T.H.).

**RNA-seq analysis.** Larval *casper* zebrafish were transplanted with ∼50 ZMEL1-GFP cells and allowed to form widespread, pigmented metastases for 18–21 days. At that point, the fish were completely disaggregated using Liberase, and a single cell suspension was made by resuspending the cells in DMEM with 2% FBS. Simultaneously, ZMEL1-GFP cells maintained in culture were trypsinized and

resuspended in DMEM/2% FBS. Both populations underwent fluorescence-acti-vated cell sorting sorting using an Aria system, and sorted directly into DMEM with 20% FBS to ensure higher cell viability. The cells were then pelleted and RNA was isolated using RNA isolation kits (Zymo Research). Because of small amount of RNA recovered from this process, all RNA samples underwent an RNA amplification step using the NuGen system, followed by Illumina library preparation and attachment of barcodes. Samples were sequenced using the HiSeq2500 with approximately 20 million reads per sample, using 50 bp single end reads. Reads from each RNA-Seq run were mapped to the zebrafish reference genome version danRer7 from the UCSC Genome Browser using GSNAP and quantified on the gene level using HTSeq and Ensembl version 75. Differential expression analysis was performed using DESeq2. The zebrafish gene symbols were mapped to their human orthologues using the DIOPT tool (http://www.flyrnai.org/cgi-bin/DRSC_orthologs.pl). Genes with a corrected *P* value of < 0.05 were considered significant.

**Gene set enrichment analysis.** Comparison of zebrafish melanoma to human melanocytes: We used previously published data in which human ES cells were induced to become either neural crest or mature melanocytes[34]. Illumina microarrays were performed on pre-differentiation human emrbyonic stem cells, day 11 neural crest cells, and day 26 mature melanocytes. From this analysis, normalized expression values were generated, and ranked gene lists for neural crest versus melanocyte (compared with day 0 cells) were input into GSEA PreRank as RNK files. From the zebrafish RNA-seq data set, we created a genelist of either upregulated (2040) or downregulated (342) genes in the disseminated ZMEL1-GFP versus *in vitro* ZMEL1-GFP cells and these were input into GSEA as GMX files. For all situation in which there were duplicate gene identifiers, we collapsed these by selecting the most highly expressed of the probeset or transcript ID. We then ran GSEA PreRank using the default weighted statistic.

Comparison of zebrafish melanoma to human melanomas: We used previously published data in which human melanomas were categorized into invasive versus proliferative subtypes[3,14]. From this data, we created a ranked list of genes in either category, which were then input into GSEA as RNK files. The zebrafish data was the same as that described above. GSEA PreRank was similarly run using the default parameters.

**Human TCGA survival analysis.** RNA-sequencing and clinical data for cutaneous melanoma patients was downloaded using the R package TCGA-assembler[35]. Subsequent survival analysis was performed in R v3.2.2. For survival analysis, patients were stratified into based on stage (with grade 3 and grade 4 patients pooled into one group). Genes were selected from the human GSEA analysis above, as well as selected groups of zebrafish genes identified from the ZFIN (http://www.zfin.org) database. Cox proportional hazard analysis was performed using the 'coxph' function in the survival package (Terry M. Therneau and Patricia M. Grambsch (2000). Modelling Survival Data: Extending the Cox Model. Springer, New York. ISBN 0-387-98784-3), where *P* values are reported for wald's test. For Kaplan–Meier analysis, patients within each grade grouping were split into low, medium, and high tertiles and survival data obtained from the cBIO portal (https://cbiologin.mskcc.org). Associations for survival were assessed with the 'survfit' and 'survdiff' functions from the survival package.

**Ingenuity pathway analysis.** From the RNA-seq analysis, the top 1,000 up or downregulated genes were used as input into IPA (http://www.ingenuity.com/ products/ipa). The Core Analysis module was selected, using Genes and Endogenous Chemicals. Only genes with a significant FDR < 0.05 were used for the analysis. For identification of likely extracellular factors causing the gene expression signature, we used both the Canonical Pathways as well as Upstream Regulator analysis modules, with the *P*-values shown in Fig. 4 derived from this analysis.

**Drug screening assay.** For convenience, we combined a 1:1 mixture of 25,000 ZMEL1-GFP along with ZMEL1 cells overexpressing a SLUG-tdTomato transgene (for an unrelated question) and seeded these in black wall, clear bottom 96-well plates (Corning, catalog #3340) in 100 μl of DMEM/10% FBS. For this manuscript, the two cells were analysed separately and only the ZMEL1-GFP cells used. The cells were allowed to adhere overnight, and then media changed to drug containing media at the indicated concentrations for each agent. These were done at either low (0.5%) or high (10%) serum conditions. Cells were grown for a total of 4 days after the addition of drug, with media/drug refreshed at day 2. At the end of the treatment, the cells were fixed with paraformaldehyde (4%), and stained using Hoechst 33342 (4 g ml$^{−1}$).

The agonists in the screen described in Fig. 3a were used at the following concentrations:

EDN1: 1.56–100 nM
EDN3: 1.56–100 nM
IGF1: 0.78–50 μM
SCF: 3.125–200 nM
LDL: 0.625–40 μM
Dopamine: 4.68–300 μM
L-DOPA: 4.68–300 μM

**High-content image analysis of ZMEL cells.** High-content image analysis was used to measure the effects of pharmacological agents on ZMEL cell proliferation and differentiation.

Using a GE INCell 6000, 2 fields were captured per well using a Nikon $10\times$ Plan Apo objective, 0.45NA. Images of the Hoechst signal was captured via the 4,6-diamidino-2-phenylindole channel with 0.1 and 0.3 seconds exposure in order to identify the nucleus and cell body respectively. The FITC channel was used for GFP with a 1 second exposure, and the dsRed channel used for tdTomato, with a 2 second exposure. Using GE Workstation software, nuclei were identified using a top-hat segmentation of the first (shortest exposure time) Hoechst image and cell bodies using a global threshold of the second (longer exposure time) Hoechst image, with identified nuclei as seed objects. GFP-positive cells were classified as having an average intensity within the nuclear mask of >500 in the GFP channel and tdTomato-positive cells with an average intensity >360 in the dsRed channel. Measurements were captured for all cells as well as GFP and tdTomato cells separately. Cell elongation is defined as the ratio of the minor to major axis of the cell. Major axis is the longest straight-line interval that can be drawn in the cell mask, while the minor axis is the longest straight line perpendicular to the major axis.

**High-content image analysis of human cells.** A 1:1 mixture of 2500 A375 cells (RFP positive) and SK-Mel28 cells (no fluorescent protein) were seeded together in 96-well plates and treated under the same conditions as the Zebrafish cells except now low serum conditions were at (0.5%) and high (5%).

Using a GE INCell 6000, 2 fields were captured per well using a Nikon $10\times$ Plan Apo objective, 0.45NA. Images of the Hoechst signal was captured via the 4,6-diamidino-2-phenylindole channel with 0.1 and 0.3 seconds exposure in order to identify the nucleus and cell body, respectively. The dsRed channel used for RFP, with a 2-second exposure. Nuclei and cell bodies were identified as per the Zebrafish cells. Cells were classified as either RFP positive (A375; having an average intensity within the nuclear mask of greater than 283 in the dsRed channel) or negative (SK-Mel28). Measurements were captured for all cells as well as RFP positive and non-fluorescent cells separately.

**Composite Z-score used in heatmaps.** Our composite Z-score is similar to the one defined in (ref. 36). The Z-score for a measured quantity (for example, the number of cells) $x_i$ in one replicate drug treated well is defined as:

$$Z_i = \frac{x_i - \mu_c}{2\sigma_c},$$

where $\mu_c$ and $\sigma_c$ is the mean and standard deviation of the measurements in the control wells with the exact same conditions as in the $i$'th replicate well except no drug is present. (In our case these are control wells containing same vehicle and on the same plate, with cells from the same reservoir as the drug treated well). Thus, each replicate well receives a signed Z-score corresponding to the number of standard deviations it falls above or below the mean of a well-defined mock-treatment distribution.

The composite Z-score is then defined as:

$$Z_{\mathrm{comp}} = \mathbf{u} \cdot \mathbf{Z}$$

where $\mathbf{u} = [u_1, u_2, ..., u_i, ..., u_n]$, (and $u_i = 1/\sqrt{n}$ for all $i$) is a vector of unit length in $R^N$ and $\mathbf{Z} = [Z_1, Z_2, ..., Z_i, ..., Z_n]$ is a vector where the components are the Z-scores for $n$ individual replicate measurements from $n$ wells.

In the limit of perfect reproducibility all replicate wells have the same Z-score ($Z_1 = Z_2 = ... = Z_i$) and $\mathbf{Z}$ and $\mathbf{u}$ are parallel. $Z_{\mathrm{comp}}$ is the length of the projection of $\mathbf{Z}$ onto the line spanned by $\mathbf{u}$ – the direction in replicate space, which represents perfect reproducibility.

**Melanin content assay.** Treated cells were grown in standard tissue culture flasks and then trypsinized. Equal numbers of cells from each condition (that is, dimethylsulphoxide versus drug treated) were pelleted at $500\,g$ for 5 min, then resuspended in 200 μl of 1N NaOH and heated at 80 °C. The pellets were vigorously pipetted to homogenize them, and then the cell extract transferred to a 96-well plate to be read at 405 nM. On each day of experiments, a standard curve of melanin using commercially available purified melanin (Invitrogen) was used as a control. Values are expressed relative to the dimethylsulphoxide control treated cells.

**Creation of EDN3b/ECE2b CRISPR mutants.** We designed multiple guide RNAs (gRNAs) against the EDN3b or ECE2b coding sequence, and synthesized these via *in vitro* transcription. The gRNAs were incubated with purified recombinant Cas9 protein (PNA Bio), and these were coinjected (75 ng μl$^{-1}$ of RNA and 500 ng μl$^{-1}$ of protein) into single cell zebrafish embryos. A portion of the injected clutch was separated at 24 hpf and genomic DNA extracted. PCR amplification of the region flanking the expected cut site was performed, followed by the T7E endonuclease assay to identify those gRNAs which gave the expected cut size. The remaining fish from that clutch were raised to adulthood, and the T7E assay repeated to identify founders capable of passing on the mutation in the germline. After identification of two such founders, these were then incrossed to yield trans-heterozygous F1 fish which were examined for phenotype of the endogenous melanocytes. Because these F1s will be a combination of WT, true heterozygotes and transheterozygotes, we only performed transplants into fish that had clearly abnormal melanocyte patterns. The control WT transplants were done into a separate group of fish that had been injected only with Cas9 protein and no gRNA and 100% of cases had normal melanocyte development. The transplants in all these animals was done identically to the *casper* transplants except that the total dose of irradiation was 20 Gy and the cell number was 300,000 cells per animal. The control, WT fish were handled in an identical manner except that they were injected with purified Cas9 protein alone with no gRNAs. The relevant sequences:

EDN3b
gRNA: 5′-GGATAAATGTACTCACTGTG-3′
F primer for genotyping: 5′-TCTCCAGTCTTCTCTGGTGGTT-3′
R primer for genotyping: 5′-GTGTGACAGCGAAAGAGTAACG-3′

ECE2b
gRNA: 5′-GGCTCCGGCTATTGAATGGT-3′
F primer for genotyping: 5′-TGACGCATATGTGAAGGTAAAAA-3′
R primer for genotyping: 5′-AACCACAGAGACAGCCATACCT-3′

**Measurement of tumour size in WT and mutants.** Adult fish were transplanted subcutaneously as described above, except that fish received either 1 or 2 transplants, since metastases are difficult to visualize in the AB background (as opposed to casper). Each fish was imaged using brightfield and GFP imaging on both the left and right side at 1–40 days post transplant (dpt). WT, EDN3b and ECE2b fish were imaged on a given day using identical exposure times and zoom. Each image was imported from the Zeiss Zen software as TIFFs and analysed in Image J. Each tumour was segmented using the default ImageJ segmentation algorithm and thresholding applied uniformly across all groups. Multiple measurement parameters were extracted, and we found highly consistent results between tumour 'area' and tumour 'perimeter', as expected. The two groups were compared with each other using unpaired ANOVA.

**Kaplan–Meier analysis.** All animals were followed for a period of 40–45 days and survival measured using the Kaplan–Meier method. The differences between the WT, EDN3b and ECE2b-recipient fish were analysed using the logrank statistic.

**Data availability.** The RNA-seq data used in this study is available via the NCBI GEO repository with accession at: http://www.ncbi.nlm.nih.gov/geo/query/acc.cgi?acc=GSE90143. Any other data is contained in this manuscript or in the Supplementary Information or available from the authors upon request. The human TCGA survival is derived from the publically available cBIO portal (https://cbiologin.mskcc.org).

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

## Acknowledgements

We would like to thank Wenjing Wu for assistance with illustrations. This work was supported by the NIH Director's New Innovator Award (DP2CA186572), K08AR055368, the Melanoma Research Alliance Young Investigator Award, an AACR/ASCO Young Investigator Award, Consano, and the Alan and Sandra Gerry Metastasis Research Initiative at the Memorial Sloan Kettering Cancer Center. S.H. was supported by the James S. McDonnell Foundation. The MSKCC High-Throughput Screening Core is supported by an NIH Core Grant (P30 CA008748).

## Author contributions

R.M.W. conceived the project. The zebrafish transplant and imaging experiments were performed by I.S.K., E.R.K., Y.Z., M.Z., K.R. and L.L. The image analysis of the zebrafish was performed by S.H., I.S.K., and J.B.X. The InCell screen and validation was performed by E.R.K., S.H., T.S.V., M.F. and R.G. The RNA-seq was analysed by W.L. The human TCGA analysis was performed by R.L.B. The histological analysis was performed by T.H. The CRISPR knockout experiments were performed by I.S.K.

## Additional information

**Competing financial interests:** The authors declare no competing financial interests.

**Publisher's note**: 

