## [Peer Review File · Nature Communications]

Reviewers' comments:

Reviewer #1 (Remarks to the Author):

This is a well-written and interesting paper that aims to describe the cellular and molecular phenotypes of melanoma cells in the metastatic context, and then identifies EDN3 as a regulator of the melanoma phenotype in the microenvironment. The big question the authors aim to address is how melanoma cells interact with the microenvironment for metastasis and growth, and they relate this to the process of phenotype switching (the process whereby melanoma cells - and other cancers - change their phenotype to undergo metastasis). Unlike the acquisition of de novo mutations, phenotypic switching involves a response to a microenvironment. Identifying what these key factors are - be they in the host or the tumour - is critical to preventing and treating metastatic cancers (Key publications include: Hoek et al., *Cancer Research* 2008; Hoek and Goding, *PCMR* 2010; O'Connell et al., 2013 *Cancer Discovery*; Saez-Ayala et al., 2013 *Cancer Cell*). EDNs are important contributors to melanocyte and melanoma development, and for tumor-host interactions (reviews: Liu, Fukunaga-Kalabis, Li and Herlyn., 2014; Saldana-Caboverde and Kos, *PCMR* 2010), and are mediators of neural crest and melanocyte fates (e.g. Mica et al., *Cell Reports* 2013).

The most important experiments are in Figure 4, where the authors make mutant lines in EDN3b and ECE2b, and transplant melanoma cells into mutant lines and see reduced tumor growth. While the data is encouraging, it is still preliminary and requires additional experimental evidence to support their claims. The biological data lacks quantification throughout the manuscript, and the data would be stronger with more than one cell line to minimize the effects of cell line variability.

1. A confusing aspect of the work is that the authors are not always consistent between addressing metastasis and/or phenotypic switching. Figure 1B provides evidence of visualizing early metastatic spread from the tumor at the primary site of injection, and this is convincing (this is similar to what the group has published in *Cancer Research*, and Tang et al., 2016 *Nature Communications* also use the transparent casper fish to visualize BRAFV600E metastasis). However, the authors do not capitalize on this in the manuscript, choosing instead to use multiple comparisons (Figure 2: cells in the petri dish versus the melanoma growths in the embryo; Figure 4 cells at the primary site of injection). This inconsistency makes it difficult to fully engage with the story that is presented.

2. In Figure 1, the authors use a zebrafish melanoma cell line, and show in two transplantation assays that the cells go from an undifferentiated to a differentiated state (i.e. become pigmented and dendritic). This is similar to the work they have published in *Cancer Research* (2015). The author acknowledge that this is what we expect with transplantation assays with mouse cell lines and patient derived xenografts, so it is not surprising that the cells differentiated once they form tumours. Figure 1C, single panels of cells suggest that that as transplanted ZMEL cells are within the larvae embryo they take on differentiation characteristics. This data must be quantified (i.e. cell number and dendricity). Again, testing with one cell line only diminishes the significance of their findings.

3. In Figure 2, tumours from the transplantation assays are compared to the cell line expression pattern by RNA-SEQ. This experiment uses the cells in culture as the controls and compares with tumours in zebrafish larvae. A stronger experiment would have been to use the tumours in adult zebrafish at the red arrow to the green mass at the injection site (Figure 1; a metastatic tumour versus the primary tumor). Instead, using cell lines in culture as the reference they compare to the larvae injection model.

4. The most exciting part of the work is the screen in the cell lines (Figure 3) that is translated to Zebra fish models (Figure 4). Using CRISPR, they generate EDN3b and ECE2b mutant lines, and then directly test how the ZMEL line grows and differentiates.

The data is encouraging, but again needs to be quantified differently. Images of the surface of the tumour is not sufficient, and given that they are interested in metastasis, it is important to find another approach that is not based on a superficial imaging of the fish (e.g. dissect tumour and count cell numbers). I actually think it is confusing to interpret the tumors at the xenograft site as metastasis - aren't they just measuring the impact of the genetic mutants on melanoma differentiation and tumorigenicity? Histology is needed so we are able to see just how the melanomas grow and invade in the host, and to validate the GFP intensity data (the tumours could just be deeper and not as easy to see). Despite what is written in the paper, it would have been better to use their casper system here, as that is the assay they have developed and actually enables the detection of metastasis in living animals.

5. The differences in survival are actually quite small - are the n numbers the same as in 4h? Have these studies been performed in matched sibling controls to compensate for background strain differences and CRISPR off-targets? Can these effects be validated by restoring EDN3 to the Zebra fish?

6. What is the expression pattern of EDN3 and ECE, and what is the evidence that these pathways directly act upon the tumor phenotype switch rather than some indirect action (e.g. vessel growth)?

7. If GFP intensity is reduced (Figure 4i), does this mean that mitf-GFP expression is reduced, and thereby that the mitf promoter is being turned off or down? How does that impact upon the expression of the BRAFV600E transgene that is expressed from the mitf promoter? It is possible that the effects they are seeing on the tumor are indirect and simply reflect reducing BRAFV600E levels.

8. The statistics for Figure 4 should be ANOVA not unpaired t-test (it says ANOVA in the Figure 4 legend, but then unpaired t-test in the methods).

9. Minor: The sentence that "Whereas melanin in mammals is not a consistent marker of differentiation, it is highly robust in zebrafish because they produce a much darker variant of melanin due to polymorphisms in the SLC24A5 gene." is not quite right. From the literature, human and zebrafish melanomas can be both pigmented and unpigmented, and the polymorphism in SLC24A5 reflects a mutation that contributes to European light skin (Lamason et al., Science 2005). Along these lines, does the pigment in the tumour interfere with the GFP analysis?

Overall, this manuscript has potential, and addresses an important question about how the environment affects phenotypic switching. The presentation of the figures is excellent, although the findings can be overstated. Significant experimental and text changes need to be made to ensure the model and idea being tested is consistent throughout the body of work, and that the data is fully validated and properly quantified.

Reviewer #2 (Remarks to the Author):

The manuscript by Kim and colleagues tackles the interesting issue of phenotype switching of melanoma cells and its role in metastasis. The study is based upon an intriguing Zebrafish model of melanoma metastasis in which the fate of individual disseminated cells can be tracked. The investigators show that metastatic cells are initially dedifferentiated and then undergo phenotype switching and reacquire the pigmented and differentiated phenotype in order to grow at new sites. A screen of potential factors involved in phenotype switching revealed a role for EDN3, SCF and IGF1, with EDN3 being the most important regulator. CRISPR/CAS technology was then used to demonstrate the importance of EDN3 in phenotype switching and metastatic growth at new sites.

Overall this is a very nice, well controlled and elegant study. It adds new insight into how melanoma cells metastasize and colonize new organs. I only have a few, mostly conceptual, questions.

1. What is the source of the EDN3? Does this come from stromal fibroblasts or other equivalent cells?
2. Is there any correlation between EDN3 levels in human melanoma patients and risk of metastasis development?
3. It would be helpful to the reader if the authors could include a summary scheme showing how EDN3/IGF1/SCF etc interaction to regulate phenotype switching, metastasis etc.

Reviewer #3 (Remarks to the Author):

This is a beautiful study on the influence of the microenvironment on phenotype switching in melanoma. The authors utilize a zebrafish model, but also include 2 human melanoma cell lines, and analyze the TCGA data for melanoma. They first show that undifferentiated cells metastasize in vivo and switch to a pigmented proliferative state. This differentiated gene signature is associated with poor survival in patients. One of the micro environmental factors is endothelin 3, a knockout zebrafish model inhibits tumor growth significantly. The influence of the micro-environment on metastasis is of high interest and the potential secreted factors identified in this study could be therapeutic targets. The methodology is very strong and the data presented robust. The data supports the conclusions.

The major points with potential for improvements are:

1. The RNAseq analysis compares in vitro cell line with in vivo metastatic lesions. It is possible that the switch to in vivo already induces many of the changes observed. It would be optimal to compare in vivo tumor cells before the phenotype switch, ie in the first week, with the later proliferating lesions. If there is not enough tissue available, nanostring for the relevant genes could be an option.
2. The survival curves for stage I and stage III/IV patients both have similar kinetics (100-200 months). Please explain in more detail how the data was collected, survival from diagnosis or biopsy? If from diagnosis, the PMEL high samples might have simply been taken at a later disease progression state and thus confound the results. Also, is there any relationship between PMEL status and metastatic site, mutational status, or other confounding factors?
3. The agonist screening data presented in Figures 3b-h is not completely convincing. Why do the authors present z-scores instead of fold change? That way it is hard to see the magnitude of the changes observed. As seen in FigH the variability is very high. Also, there is no clear dose response relationship in the A375 cells treated with EDN1/3. A very low concentration of 1.5nM already induces significant proliferation and elongation?
4. How can the induction of pigmentation/ differentiation in ZMEL1 cells by L-DOPA, but lack of proliferation induction be explained?
5. The authors do not show true metastatic lesions in the EDN3 CRISPR fish. Using the GFP signal, this should be possible to do even in the AB strain. Alternately, the intra-vasal inoculation in larvae could be used in these backgrounds. Showing s.c. injected tumors as surrogate for metastasis is a good indication, but it leaves the possibility that EDN3 loss in the microenvironment inhibits tumor cell implantation at the metastasis sites and not the phenotype switch to a proliferative state.

Minor points:

1. Move references 1-3 from abstract to introduction.
2. The methods for the screen refer to Fig4 instead of 3.

Reviewer #1 (Remarks to the Author):

This is a well-written and interesting paper that aims to describe the cellular and molecular phenotypes of melanoma cells in the metastatic context, and then identifies EDN3 as a regulator of the melanoma phenotype in the microenvironment. The big question the authors aim to address is how melanoma cells interact with the microenvironment for metastasis and growth, and they relate this to the process of phenotype switching (the process whereby melanoma cells - and other cancers - change their phenotype to undergo metastasis). Unlike the acquisition of de novo mutations, phenotypic switching involves a response to a microenvironment. Identifying what these key factors are - be they in the host or the tumour - is critical to preventing and treating metastatic cancers (Key publications include: Hoek et al., Cancer Research 2008; Hoek and Goding, PCMR 2010; O'Connell et al., 2013 Cancer Discovery; Saez-Ayala et al., 2013 Cancer Cell). EDNs are important contributors to melanocyte and melanoma development, and for tumor-host interactions (reviews: Liu, Fukunaga-Kalabis, Li and Herlyn., 2014; Saldana-Caboverde and Kos, PCMR 2010), and are mediators of neural crest and melanocyte fates (e.g. Mica et al., Cell Reports 2013).

The most important experiments are in Figure 4, where the authors make mutant lines in EDN3b and ECE2b, and transplant melanoma cells into mutant lines and see reduced tumor growth. While the data is encouraging, it is still preliminary and requires additional experimental evidence to support their claims. The biological data lacks quantification throughout the manuscript, and the data would be stronger with more than one cell line to minimize the effects of cell line variability.

1. A confusing aspect of the work is that the authors are not always consistent between addressing metastasis and/or phenotypic switching. Figure 1B provides evidence of visualizing early metastatic spread from the tumor at the primary site of injection, and this is convincing (this is similar to what the group has published in Cancer Research, and Tang et al., 2016 Nature Communications also use the transparent casper fish to visualize BRAFV600E metastasis). However, the authors do not capitalize on this in the manuscript, choosing instead to use multiple comparisons (Figure 2: cells in the petri dish versus the melanoma growths in the embryo; Figure 4 cells at the primary site of injection). This inconsistency makes it difficult to fully engage with the story that is presented.

We agree that we did not use consistent terminology throughout the manuscript. We have clarified the text to indicate that our study is best able to address the biology of phenotype switching in subcutaneous (as opposed to visceral) metastasis. This pattern of subcutaneous spread is one of the dominant sites of metastatic disease in humans, being the site of primary spread in 20% and occurring in 50% of patients with visceral spread (Meier, et al, Br J Dermatol. 2002 Jul;147(1):62-70). These patients are classified as stage IV (Tx/Nx/M1a), similar to what is seen here:

In the adult transplantation model (Figure 1a, top), our primary site of injection is in the skin, but when the tumors leave the primary injection site and arise in new locations, they most typically arise in subcutaneous sites, analogous to subcutaneous spread in human patients. Similarly, in the embryo transplant model (Figure 1b, bottom) we also see a strong tropism for subcutaneous sites and the majority of the visible tumor mass is composed of subcutaneous growths which also become pigmented. For these reasons, we feel that both the adult transplant studies as well as the embryo transplant studies particularly address subcutaneous metastases, and the bulk of our downstream analysis focuses on phenotype switching in these subcutaneous sites. We have modified the text surrounding Figure 1 to indicate that we primarily focused on subcutaneous metastases in the observations of phenotype switching.

Given this, in Figure 4 our logic was along the same lines – we were primarily focusing on subcutaneous tumor metastases, and utilized the multiple subcutaneous injection procedure to ensure that our findings were not isolated to a single particular anatomic location, and best allowed us to quantify the effects at specific subcutaneous sites. We felt it important to perform these studies in a wild-type (AB) background for two reasons. First, to ensure that our findings have the most relevance to human patients, we wanted a situation in which the resident endogenous melanocytes are still present, since that would be the case in patients. Second, because CRISPRs produce an allelic series of mutations we wanted to use animals in which we could be certain that the CRISPR was exerting a biological consequence on the endogenous normal melanocytes, and we could take advantage of known literature that loss of EDN3 or ECE2 function should lead to defects in endogenous melanocytes (as shown in Figure 4a-c).

To confirm that performing these experiments was independent of the strain, we have now performed an additional series of transplants using the casper strain, in an identical fashion to what we did for the AB animals and found largely similar results. In this smaller series of animals, we found that the EDN3b (n=6) recipients had significantly smaller tumor areas compared to WT (n=5) recipients (area=1.6 vs. 0.95, $p<0.05$, ANOVA), analogous to what we saw in the AB strain. We found only a trend towards a decrease in tumor area in the ECE2b recipients in the casper strain (area=1.6 in WT vs. 1.4 in ECE2b, $p>0.05$, ANOVA). We suspect this discrepancy is due to the significantly smaller numbers in this study as compared to the AB strain experiments. All of these results have been added to the manuscript text and included as Extended Data 9.

2. In Figure 1, the authors use a zebrafish melanoma cell line, and show in two transplantation assays that the cells go from an undifferentiated to a differentiated state (i.e. become pigmented and dendritic). This is similar to the work they have published in Cancer Research (2015). The author acknowledge that this is what we expect with transplantation assays with mouse cell lines and patient derived xenografts, so it is not surprising that the cells differentiated once they form tumours. Figure 1C, single panels of cells suggest that that as transplanted ZMEL cells are within the larvae embryo they take on differentiation characteristics. This data must be quantified (i.e. cell number and dendricity). Again, testing with one cell line only diminishes the significance of their findings.

To quantify the degree of dendricity after transplantation, we created a new ZMEL1 melanoma cell line in which we included a palmitylated tdTomato cassette, which is membrane bound and much more clearly labels the dendrites when compared the cytoplasmic GFP. We then transplanted these cells into a series of n=10 fish using the embryo transplant assay, and then used confocal imaging to image the cells either on the day of transplant versus 5 days post transplant. Of the 10 fish, 7 survived throughout the entire imaging period and were included for analysis. On the day 1 fish, we counted 43 individual cells and found that 0% were dendritic by this measure. In contrast, by day 5, we counted 41 cells and found that now 83% had evidence of dendricity. We have incorporated this data into the body of the manuscript and included it as Extended Data 1.

We also performed transplants with an additional melanoma cell line called ZCREST1. This line also harbors a BRAF^{V600E} mutation and p53 loss of function, but instead the GFP reporter gene is driven by our previously described crestin neural crest promoter (Kaufman, et al, Science. 2016 Jan 29;351(6272)). We then performed adult transplants in a series of 5 fish (4 of which survived) using the same methods we used for the

ZMEL1 line. Identical to what we see with the ZMEL1 line, all of the tumors start out unpigmented but all 4 fish showed extensive pigmentation in both the primary as well as metastatic subcutaneous sites by days 14-44. We have included this data in Extended Data 2.

3. In Figure 2, tumours from the transplantation assays are compared to the cell line expression pattern by RNA-SEQ. This experiment uses the cells in culture as the controls and compares with tumours in zebrafish larvae. A stronger experiment would have been to use the tumours in adult zebrafish at the red arrow to the green mass at the injection site (Figure 1; a metastatic tumour versus the primary tumor). Instead, using cell lines in culture as the reference they compare to the larvae injection model.

We agree with this limitation of our study and have attempted, in a multitude of ways, to perform RNA-seq on cells at the primary skin transplant site versus secondary cutaneous metastases. The major issue is that the secondary subcutaneous metastases are extremely small (often on the order of 100 cells), making RNA-isolation from this number of cells very difficult. We have tried multiple digestion procedures (Liberase, Trypsin, Collagenase), FACS parameters (low-pressure sort, FACS-Aria vs. MoFlow machines) and RNA isolation kits (Zymo, Qiagen) but were never able to obtain sufficient high quality RNA from the secondary subcutaneous metastases to perform RNA-seq. An example of one such sort is shown here, demonstrating low quality, low amounts of RNA from small numbers of cells:

Plate Barcode	Well	Tube Barcode	Client Sample ID	Concentration (ng/ μ L)	Mass (μ g)	RQN/GQN	Comment
MTR-00962	D1	161424270	post-sort 1	too low	0.1	7	Qubit cannot detect any mass, mass calculated from BA. Degraded material present.
MTR-00962	E1	161424148	post-sort 2	too low	0.01	2.4	Qubit cannot detect any mass, mass calculated from BA. Sample entirely degraded.

While comparing the cells in vitro to the cells after transplantation is not ideal, the bulk of the cells we sorted from the embryos were from subcutaneous metastases. Given the above technical limitations, this made it the most practical solution to our ultimate goal of assessing the gene expression changes that are associated with phenotype switching in subcutaneous metastases. Future studies which take advantage of low or even single cell input RNA-seq will be able to address the gene expression changes in primary injection site versus secondary sites. We realize that this is still a technical hurdle for our field, and while we think it is beyond the scope of this manuscript, this is a very important area for future studies.

4. The most exciting part of the work is the screen in the cell lines (Figure 3) that is translated to Zebra fish models (Figure 4). Using CRISPR, they generate EDN3b and ECE2b mutant lines, and then directly test how the ZMEL line grows and differentiates.

The data is encouraging, but again needs to be quantified differently. Images of the surface of the tumour is not sufficient, and given that they are interested in metastasis, it is important to find another approach that is not based on a superficial imaging of the fish (e.g. dissect tumour and count cell numbers). I actually think it is confusing to interpret the tumors at the xenograft site as metastasis - aren't they just measuring the impact of the genetic mutants on melanoma differentiation and tumorigenesis? Histology is needed so we are able to see just how the melanomas grow and invade in the host, and to validate the GFP intensity data (the tumours could just be deeper and not as easy to see). Despite what is written in the paper, it would have been better to use their casper system here, as that is the assay they have developed and actually enables the detection of metastasis in living animals.

As described above, we primarily used the multiple subcutaneous injections in the wild-type fish as a means of simulating subcutaneous metastases, which is what our casper models most easily recapitulate. We have included the data from the casper transplants in Extended Data 9.

To quantify the tumor effects in the EDN3b or ECE2b backgrounds, we have performed histologic sectioning of 12 tumors from 6 different fish and performed 6 different histological/IHC stains. This data has

been added as a new Figure 5. The stains included: 1) H&E, 2) phospho-H3 as a marker of proliferation, 3) cleaved caspase as a marker of apoptosis, 4) Fontana-Mason as a marker of melanin pigmentation, 5) GFP as a marker of transgene expression from the *mitf* promoter, and 6) BRAF^{V600E} as a marker of transgene expression from the *mitf* promoter. We collaborated with an expert in dermatopathology (Travis Hollmann) who performs similar studies in human melanoma patients and has expertise in reading these types of slides.

By H&E, 4/4 tumors in the EDN3b and ECE2b fish had evidence of central necrosis, compared to 0/4 of the WT recipients (Figure 5). This is accompanied by a significant increase in caspase staining (Figure 5). This likely explains why the tumors in these two mutant backgrounds are smaller than the WT background. We saw a modest difference in pH3 staining in the EDN3b recipient, but not in the ECE2b recipients, which may relate to the more direct effect of abrogating the ligand instead of only one of the processing enzymes. This discrepancy will need further exploration in future studies. We measured the number of Fontana-Mason positive cells, and found a significant decrease in the number of pigmented cells in both the EDN3b and ECE2b recipients compared to WT. We found no difference in the percentage of either BRAF or GFP positive cells – virtually all tumor cells within the field were positive for both markers, indicating that the decrease in GFP intensity shown in Figure 4i is related to overall tumor size and not a decrease in expression of the transgenes. All of these results have been added into the body of the manuscript.

5. The differences in survival are actually quite small - are the n numbers the same as in 4h? Have these studies been performed in matched sibling controls to compensate for background strain differences and CRISPR off-targets? Can these effects be validated by restoring EDN3 to the Zebra fish?

The animal numbers are indicated in the legend for Figure 4 and in the text, and the survival curves are for the same numbers of fish in Figures 4h-j. Overall, at day 50, we found that survival in the WT background was 6.25%, whereas it was 54.1% in EDN3b and 42.1% in ECE2b. We feel this is a meaningful increase in survival considering we changed just a single microenvironmental factor. All of our studies were done in sibling matched controls.

Although rescue of the effects we see would be ideal, we know of no straightforward way of performing these studies in the zebrafish other than to create a new heat shock or other form of inducible EDN3 transgenic line. Systemic delivery of endothelin peptides is not feasible given their short half-life and inconsistent cross-species effects after IV injection (i.e. available EDN3 peptides are only of human origin). Because the creation and validation of new transgenic zebrafish lines takes anywhere from 6-12 months, we do not feel that this is a viable approach for this manuscript but it would be a highly valuable experiment for later studies.

6. What is the expression pattern of EDN3 and ECE, and what is the evidence that these pathways directly act upon the tumor phenotype switch rather than some indirect action (e.g. vessel growth)?

Recently published work (Krauss, et al, Biol Open. 2014 May 23;3(6):503-9) demonstrates that EDN3 is broadly expressed in the zebrafish epidermis, which is consistent with other work from mouse studies showing that endothelins are derived from skin keratinocytes (Imokawa, et al, J Biol Chem. 1992 Dec 5;267(34):24675-80) and that overexpression of EDN3 from the keratinocyte promoter is sufficient to cause hyperplasia of distal melanocytes (Garcia, et al, J Invest Dermatol. 2008 Jan;128(1):131-42). In our zebrafish, we have attempted in situ hybridization for ECE2b, but the expression appears quite broad and we could not clearly identify its cell of origin. However, previous work in normal human keratinocytes showed evidence of ECE-dependent endothelin production (Pernet, et al, Exp Dermatol. 2000 Dec;9(6):401-6), suggesting this is likely at least one origin of endothelin. We have modified the text to indicate that the source of EDN3 is likely the surrounding epidermal keratinocytes, but the precise localization of ECE2b awaits further study.

Given our histological findings above, with central necrosis, it seems likely that the effects we are seeing in our subcutaneous tumors is due to a combination of both direct effects (mediated by binding of EDN3 to the EDNRB receptor on melanoma cells) along with an effect on the vasculature (due to the known expression of endothelin receptors of vascular endothelial cells). We have modified the text to reflect these two possibilities, which are not mutually exclusive.

7. If GFP intensity is reduced (Figure 4i), does this mean that mitf-GFP expression is reduced, and thereby that the mitf promoter is being turned off or down? How does that impact upon the expression of the BRAFV600E transgene that is expressed from the mitf promoter? It is possible that the effects they are seeing on the tumor are indirect and simply reflect reducing BRAFV600E levels.

As addressed above, we have performed BRAF^{V600E} and GFP IHC and see no differences in staining intensity comparing WT to EDN3b to ECE2b.

8. The statistics for Figure 4 should be ANOVA not unpaired t-test (it says ANOVA in the Figure 4 legend, but then unpaired t-test in the methods).

This has been corrected.

9. Minor: The sentence that "Whereas melanin in mammals is not a consistent marker of differentiation, it is highly robust in zebrafish because they produce a much darker variant of melanin due to polymorphisms in the SLC24A5 gene." is not quite right. From the literature, human and zebrafish melanomas can be both pigmented and unpigmented, and the polymorphism in SLC24A5 reflects a mutation that contributes to European light skin (Lamason et al., Science 2005). Along these lines, does the pigment in the tumour interfere with the GFP analysis?

We have clarified this point. What we meant to indicate is that pigmentation appears as a spectrum, and in humans melanomas are nearly universally found only in those with the European/Caucasian SLC24A5 variant, which makes a relatively light pigment compared to the African SLC24A5 variant which is much darker. Thus, the pigmentation we see in human cutaneous melanoma is still considerably lighter than what is seen in the zebrafish melanomas, since our fish all carry the darker African SLC24A5 variant.

When the tumors become extremely pigmented, we do occasionally see situations in which it can interfere with GFP imaging, but the tumor cells are still almost always visible along the edges of the cells since melanin tends to be perinuclear.

Overall, this manuscript has potential, and addresses an important question about how the environment affects phenotypic switching. The presentation of the figures is excellent, although the findings can be overstated. Significant experimental and text changes need to be made to ensure the model and idea being tested is consistent throughout the body of work, and that the data is fully validated and properly quantified.

Reviewer #2 (Remarks to the Author):

The manuscript by Kim and colleagues tackles the interesting issue of phenotype switching of melanoma cells and its role in metastasis. The study is based upon an intriguing Zebrafish model of melanoma metastasis in which the fate of individual disseminated cells can be tracked. The investigators show that metastatic cells are initially dedifferentiated and then undergo phenotype switching and reacquire the pigmented and differentiated phenotype in order to grow at new sites. A screen of potential factors involved in phenotype switching revealed a role for EDN3, SCF and IGF1, with EDN3 being the most important regulator. CRISPR/CAS technology was then used to demonstrate the importance of EDN3 in phenotype switching and metastatic growth at new sites. Overall this is a very nice, well controlled and elegant study. It adds new insight into how melanoma cells metastasize and colonize new organs. I only have a few, mostly conceptual, questions.

1. What is the source of the EDN3? Does this come from stromal fibroblasts or other equivalent cells?

Recently published work (Krauss, et al, Biol Open. 2014 May 23;3(6):503-9) demonstrates that EDN3 is broadly expressed in the zebrafish epidermis, which is consistent with other work from mouse studies showing that endothelins are derived from skin keratinocytes (Imokawa, et al, J Biol Chem. 1992 Dec 5;267(34):24675-80) and that overexpression of EDN3 from the keratinocyte promoter is sufficient to cause hyperplasia of distal melanocytes (Garcia, et al, J Invest Dermatol. 2008 Jan;128(1):131-42). In our zebrafish, we have attempted in situ hybridization for ECE2b, but the expression appears quite broad and we could not clearly identify its cell of origin. However, previous work in normal human keratinocytes showed evidence of ECE-dependent endothelin production (Pernet, et al, Exp Dermatol. 2000 Dec;9(6):401-6), suggesting this is likely at least one origin of endothelin. We have modified the text to indicate that the source of EDN3 is likely the surrounding epidermal keratinocytes, but the precise localization of ECE2b awaits further study.

2. Is there any correlation between EDN3 levels in human melanoma patients and risk of metastasis development?

The association is stronger for ECE2 rather than EDN3 itself. The data in Figure 2e/f clarify this point. For patients who present with Stage I disease (Figure 2e), there is no clear association with expression of either EDN3 or ECE2. However, for those patients who present with Stage III/IV disease, higher expression of ECE2 (but not EDN3) is associated with a significantly worse survival than those with lower levels of ECE2. The same is found for expression of the EDN3 receptor EDNRB which is present on the melanoma cells. Taken together, this data suggests that once a patient has progressed to advanced disease, the endothelin axis promotes worse survival, rather than being an initiator of metastasis itself.

3. It would be helpful to the reader if the authors could include a summary scheme showing how EDN3/IGF1/SCF etc interaction to regulate phenotype switching, metastasis etc.

This has now been included as a new Figure 6.

Reviewer #3 (Remarks to the Author):

This is a beautiful study on the influence of the microenvironment on phenotype switching in melanoma. The authors utilize a zebrafish model, but also include 2 human melanoma cell lines, and analyze the TCGA data for melanoma. They first show that undifferentiated cells metastasize in vivo and switch to a pigmented proliferative state. This differentiated gene signature is associated with poor survival in patients. One of the micro environmental factors is endothelin 3, a knockout zebrafish model inhibits tumor growth significantly. The influence of the micro-environment on metastasis is of high interest and the potential secreted factors identified in this study could be therapeutic targets. The methodology is very strong and the data presented robust. The data supports the conclusions.

The major points with potential for improvements are:

1. The RNAseq analysis compares in vitro cell line with in vivo metastatic lesions. It is possible that the switch to in vivo already induces many of the changes observed. It would be optimal to compare in vivo tumor cells before the phenotype switch, ie in the first week, with the later proliferating lesions. If there is not enough tissue available, nanostring for the relevant genes could be an option.

We agree with this limitation of our study and have attempted, in a multitude of ways, to perform RNA-seq on cells at day 5 post-transplant to those at day 30 post-transplant. The major issue is that isolating small numbers of GFP+ cells from the 5 day fish has proved extremely difficult, making RNA-isolation from this number of cells very difficult. We have tried multiple digestion procedures (Liberase, Trypsin, Collagenase), multiple FACS parameters (low-pressure sort, FACS-Aria vs. MoFlow machines) and RNA isolation kits (Zymo, Qiagen) but were never able to obtain sufficient high quality RNA to perform RNA-seq. For example, we tried to sort out cells by FACS and then isolate RNA, but this demonstrated low quality, low amounts of RNA from small numbers of cells:

Plate Barcode	Well	Tube Barcode	Client Sample ID	Concentration (ng/ μ L)	Mass (μ g)	RQN/GQN	Comment
MTR-01224	B1	196758803	post-sort 4	too low	0.004	6.9	Qubit cannot detect any mass, mass calculated from BA. Degraded material present.

While we agree that comparing the cells in vitro to the cells after transplantation is not ideal, the bulk of the cells we sorted from the embryos were from subcutaneous metastases, and felt it was the most practical solution to assessing the molecular nature of phenotype switching, at least in subcutaneous tumor growth. Advances in extremely low input RNA-seq or even single cell RNA-seq would be useful here, but even techniques such as Nanostring still require ~100ng of high-quality input RNA (<http://www.nanostring.com/support/FAQs>). We realize that this is still a technical hurdle for our field, and while we think it is beyond the scope of this manuscript, this is a very important area for future studies.

2. The survival curves for stage I and stage III/IV patients both have similar kinetics (100-200 months). Please explain in more detail how the data was collected, survival from diagnosis or biopsy? If from diagnosis, the PMEL high samples might have simply been taken at a later disease progression state and thus confound the results. Also, is there any relationship between PMEL status and metastatic site, mutational status, or other confounding factors?

The data that was included in Figure 2 are derived from the publically available TCGA dataset for melanoma, as reflected in the consortium publication (Cell. 2015 Jun 18;161(7):1681-96) and accessed via the cBio portal (<https://cbiologin.mskcc.org>). According to their data, 80% of the samples were derived from metastatic tissue, and only 20% were derived from primary tumors. According to the TCGA: "Compared to most solid tumors, primary melanomas are generally small at diagnosis; and in routine clinical practice, most or all of primary tumor tissue is used for diagnostic evaluation and is not available for molecular analyses."

Within the TCGA dataset, "Stage I" refers to patients who initially presented with localized stage I disease, yet the vast majority of those patients eventually went on to develop metastatic disease (which is when they were sampled). Similarly, "Stage III/IV" refers to patients who presented with advanced disease, and again nearly all of

those samples are of metastatic sites. The survival we used is therefore survival from the time of sample collection.

In our analysis, we utilized the TCGA nomenclature in this manner, which explains why the kinetics of the two curves are similar: nearly all patients in the study eventually developed metastases, whether they initially presented as stage I or as stage III/IV. What our analysis is most able to show is that for patients who **present** at stage III/IV, the expression of differentiation genes like PMEL and TYR portends a significantly worse prognosis. Put another way, PMEL/TYR helps to stratify metastatic patients. It is possible that PMEL could also stratify patients who initially presented at stage I but the TCGA study was not powered to detect this. Our analysis is similar to what was found in the TCGA publication, which also noted a worse prognosis for the MITF target gene group (i.e. PMEL and TYR). We did not find any other clear factors which correlated with PMEL status such as BRAF/NRAS status or location.

We have modified the text of the manuscript to reflect this information to make it more clear how we calculated survival time, and that it is best used to sub-stratify stage III/IV patients.

3. The agonist screening data presented in Figures 3b-h is not completely convincing. Why do the authors present z-scores instead of fold change? That way it is hard to see the magnitude of the changes observed. As seen in FigH the variability is very high. Also, there is no clear dose response relationship in the A375 cells treated with EDN1/3. A very low concentration of 1.5nM already induces significant proliferation and elongation?

Our decision to use z-scores is in line with guidelines published by the Broad Institute for large-scale chemical screening endeavors, and we used the same calculation published by the Broad group (Seiler, et al, Nucleic Acids Res. 2008 Jan;36). We have tried various color schemes on the z-score heatmaps to try make this more visible, but for the elongation analysis in particular we find that the effect is simply less in the ZMEL1 line compared to the A375 line, and the colors accurately reflect the magnitude of these effects. Because we too were concerned by the small magnitude of the effect, that is why we used an orthogonal assay of directly measuring melanin content (Figure 3f/g) which much more obviously confirmed the differentiation effect.

When we began screening, we were not sure what dose ranges to use, because we were testing both human and zebrafish cells and it wasn't clear whether the same dose ranges would work across species. We tried to be guided by the literature, but as it turned out, even very low doses of EDN3 (in the nanomolar range) cause the effects we see. This is highly consistent with other published literature showing nanomolar effects of endothelin peptides in vitro (Yohn, et al, Biochem Biophys Res Commun. 1994 May 30;201(1):449-57). Although in retrospect we could have started with subnanomolar doses, we did not know this a priori and this likely explains the lack of a clear linear dose response relationship. We do not fully know the source of the variable responses we see, as shown in Figure 3h, but we suspect most of this comes from imperfect imaging on the InCell 6000 system. We met with GE (the manufacturers of the InCell), but they felt that this was as optimal as we were able to achieve. It is possible that further optimization of the imaging algorithms would improve this analysis, and we have spoken with the software engineers at GE but they do not yet have a solution to reduce this variability.

4. How can the induction of pigmentation/ differentiation in ZMEL1 cells by L-DOPA, but lack of proliferation induction be explained?

We believe there are two potential explanations: 1) L-DOPA or dopamine may bind directly to a dopamine receptor that induces cAMP, which mainly will only cause an increase in pigmentation/differentiation as previously described for melanocytes. In our profiling of zebrafish melanoma cells, we did see upregulation of the DRD1 and DRD5 receptors but this was not conserved in the human samples. Nonetheless, we tested several DRD1/DRD5 receptor antagonists to see if they could rescue the pigmentation effect of L-DOPA but this was not convincing. Whether melanoma cells express dopamine receptors has been the source of considerable

controversy, with some studies supporting expression (Krummel, et al Cancer. 1982 Mar 15;49(6):1178-84) and others refuting it (Boni, et al, Melanoma Res. 1997 Apr;7(2):117-9). Taken together, it suggests to us that a non-receptor mediated effect may be at play. 2) L-DOPA may also be internalized into the melanocyte, as has been described for other cell types such as neurons, and used directly in the melanin synthesis pathway. L-DOPA is the substrate for the tyrosinase enzyme, the TYR gene product. This gives rise to the intermediate dopaquinone which is then used for subsequent melanin generation. Terminal differentiation of melanocytes is typically associated with proliferation arrest, so based on the fact that we see high levels of TYR in our disseminated cells, we think this is the more likely explanation. We have reflected the text to address these two possibilities, but recognize that this is still an open mechanistic question.

5. The authors do not show true metastatic lesions in the EDN3 CRISPR fish. Using the GFP signal, this should be possible to do even in the AB strain. Alternately, the intra-vasal inoculation in larvae could be used in these backgrounds. Showing s.c. injected tumors as surrogate for metastasis is a good indication, but it leaves the possibility that EDN3 loss in the microenvironment inhibits tumor cell implantation at the metastasis sites and not the phenotype switch to a proliferative state.

We agree that the multiple subcutaneous injections limited our ability to discern implantation failure versus phenotype switch at metastatic sites. We initially did perform these studies with the goal of specifically imaging the secondary subcutaneous metastases in the WT vs. EDN3b backgrounds, but found that the very small size and number of secondary lesions in the mutant precluded accurate imaging, and we did not feel we could confidently assess phenotype switching in that experiment. We reasoned that the multiple subcutaneous injections would be a reasonable proxy (and give us much more reliable imaging) based on the fact that metastases in both the adult (Figure 1a, top) and embryo (Figure 1a, bottom) zebrafish assays are dominated by subcutaneous deposits. This pattern of distant subcutaneous spread is one of the major sites of metastatic disease in humans, being the site of primary spread in 20% and occurring in 50% of patients with visceral spread (Meier, et al, Br J Dermatol. 2002 Jul;147(1):62-70). These patients are classified as stage IV (Tx/Nx/M1a), and we felt that our study is best capable of addressing phenotype switching in subcutaneous (not visceral) sites. It was for these reasons that we chose to use multiple subcutaneous transplants for Figure 4.

To fully address whether the loss of EDN3b or ECE2b lead to an abrogation of phenotype switching, we have performed histologic sectioning of 12 tumors from 6 different fish and performed 6 different histological/IHC stains. This data has been added as a new Figure 5. The stains included: 1) H&E, 2) phospho-H3 as a marker of proliferation, 3) cleaved caspase as a marker of apoptosis, 4) Fontana-Mason as a marker of melanin pigmentation, 5) GFP as a marker of transgene expression from the mitf promoter, and 6) BRAF^{V600E} as a marker of transgene expression from the mitf promoter. We collaborated with an expert in dermatopathology (Travis Hollmann) who performs similar studies in human melanoma patients and has expertise in reading these types of slides. This analysis showed a significant increase in cleaved caspase in both EDN3b and ECE2b, as well as a decrease in pH3 in the EDN3b background. Moreover, we measured pigmentation in the sections using Fontana-Mason, and found a significant decrease in both EDN3b and ECE2b. Taken together, this data supports a model in which loss of EDN3b from the microenvironment prevents acquisition of the differentiated/proliferative state, at least in subcutaneous deposits.

Minor points:

1. Move references 1-3 from abstract to introduction.

This has been changed.

2. The methods for the screen refer to Fig4 instead of 3.

This has been corrected.

REVIEWERS' COMMENTS:

Reviewer #1 (Remarks to the Author):

The authors have addressed my comments.

Reviewer #2 (Remarks to the Author):

The authors have addressed the points raised in the initial review.

Reviewer #3 (Remarks to the Author):

The manuscript compares the in vitro melanoma cell line to cells disseminated in vivo at d21. In their rebuttal the authors agree that this is not an ideal comparison but that isolating early in vivo lesions before widespread dissemination was too difficult and they did not succeed in collecting enough high quality RNA for RNAseq. In my opinion this would be possible if enough biological replicates would be pooled, but agree with the authors that this may be beyond the scope of the current work. Also, the authors went on to successfully validate the findings from the RNAseq screening using gene editing.

All other comments were addressed adequately and the manuscript revised accordingly.